# Isolation and Characterization of Ruminal Yeast Strain with Probiotic Potential and Its Effects on Growth Performance, Nutrients Digestibility, Rumen Fermentation and Microbiota of Hu Sheep

**DOI:** 10.3390/jof8121260

**Published:** 2022-11-29

**Authors:** Yao Wang, Zihao Li, Wei Jin, Shengyong Mao

**Affiliations:** 1Ruminant Nutrition and Feed Engineering Technology Research Center, College of Animal Science and Technology, Nanjing Agricultural University, Nanjing 210095, China; 2Laboratory of Gastrointestinal Microbiology, National Center for International Research on Animal Gut Nutrition, College of Animal Science and Technology, Nanjing Agricultural University, Nanjing 210095, China

**Keywords:** *Pichia kudriavzevii*, fiber degradation, rumen fermentation, microbial community composition, rumen native yeast

## Abstract

Yeast strains are widely used in ruminant production. However, knowledge about the effects of rumen native yeasts on ruminants is limited. Therefore, this study aimed to obtain a rumen native yeast isolate and investigate its effects on growth performance, nutrient digestibility, rumen fermentation and microbiota in Hu sheep. Yeasts were isolated by picking up colonies from agar plates, and identified by sequencing the ITS sequences. One isolate belonging to *Pichia kudriavzevii* had the highest optical density among these isolates obtained. This isolate was prepared to perform an animal feeding trial. A randomized block design was used for the animal trial. Sixteen Hu sheep were randomly assigned to the control (CON, fed basal diet, *n* = 8) and treatment group (LPK, fed basal diet plus *P. kudriavzevii*, CFU = 8 × 10^9^ head/d, *n* = 8). Sheep were housed individually and treated for 4 weeks. Compared to CON, LPK increased final body weight, nutrient digestibility and rumen acetate concentration and acetate-to-propionate ratio in sheep. The results of Illumina MiSeq PE 300 sequencing showed that LPK increased the relative abundance of lipolytic bacteria (*Anaerovibrio* spp. and *Pseudomonas* spp.) and probiotic bacteria (*Faecalibacterium* spp. and *Bifidobacterium* spp.). For rumen eukaryotes, LPK increased the genera associated with fiber degradation, including protozoan *Polyplastron* and fungus *Pichia*. Our results discovered that rumen native yeast isolate *P. kudriavzevii* might promote the digestion of fibers and lipids by modulating specific microbial populations with enhancing acetate-type fermentation.

## 1. Introduction

The rumen contains a large number of microorganisms that ferment structural and non-structural carbohydrates to produce volatile fatty acids (VFAs) and synthesize microbial proteins for the host [1]. Microbial degradation and fermentation, which can enhance ruminal digestion and absorption capacity and nutrient metabolism, are of central importance in ruminant nutrition [2]. Even if ruminal microorganisms are inherently efficient, probiotics are often added during production to further improve rumen digestion, thus increasing economic benefit.

Yeast products are widely used as probiotic additives in ruminant production [3]. Previous studies have reported that live yeast can consume ruminal oxygen and decrease the redox potential, which favors the activity of anaerobic microorganisms [3,4]. Moreover, the addition of yeast provides nutrients (vitamins, peptides and growth factors) for the host, thereby stimulating the growth of ruminal microorganisms [5]. The regulation of yeast on ruminal microorganisms is reflected in increasing fiber-degrading and lactate-utilizing bacteria, which improves further rumen fermentation efficiency and subsequent production capacity [6,7]. In ruminant production, *Saccharomyces cerevisiae* is the most commonly used yeast additive [8]. However, Ishaq et al. reported no change in the proportion of *S. cerevisiae* in the rumen after feeding active *S. cerevisiae* to dairy cows, indicating that live *S. cerevisiae* may not serve a functional purpose [9]. Therefore, the effectiveness of probiotics depends on their adaptability to a specific ecosystem.

In this regard, potential probiotic yeasts can be selected from the rumen to enhance ruminants’ adaptability [10]. The proportion of yeasts in the rumen may be more than 1 × 10^3^ CFU/mL, which plays an important role in improving rumen functions [11]. In particular, *Pichia kudriavzevii*, as a rumen native yeast, accounts for the high relative abundance of ruminal yeasts [12,13]. Results in vitro experiments have revealed that, unlike *S. cerevisiae*, *Pichia kudriavzevii* survives well in the rumen environment [14,15]. Additionally, it possesses fiber-degrading enzymes and the ability to utilize lactate. Feeding rumen-derived *P. kudriavzevii* to dairy cows has recently been applied to increase milk production and feed conversion rates [16,17]. However, the previous studies only focused on milk performance, rumen bacteria and fungi in cows [16,17]. To date, the effects of rumen native *P. kudriavzevii* on growth performance, ruminal microorganisms (bacteria and eukaryote) and the fermentation parameters and digestibility of nutrients remain poorly understood in Hu sheep.

Herein, we isolated rumen native *P. kudriavzevii* from Hu sheep to investigate its effect on growth performance, ruminal fermentation, rumen bacteria and eukaryote community and digestibility of nutrients in Hu sheep. Our study brought new insights to the application of rumen-native yeast in future sheep production practice.

## 2. Materials and Methods

The protocols of this trial and the use of animals were approved by the Animal Care and Use Committee of Nanjing Agricultural University.

### 2.1. Isolating, Identifying and Characterizing Yeast Isolates

Fresh rumen fluid samples were collected to isolate yeasts from the rumen fistula of Hu sheep. The rumen fluid was 10 times gradually diluted from 10^−1^ to 10^−5^ with 0.75% sterile saline. About 200 μL of each of the dilutions was spread on yeast agar plates consisting (1000 mL) of yeast extract 5 g, peptone 10 g, glucose 20 g, and agar 14 g. Streptomycin sulfate (2000 U/mL) and Penicillin potassium (1600 U/mL) used to inhibit bacteria were purchased from Jilin Huamu Animal Health Products Co., Ltd. (Changchun, China). These agar plates were incubated at 30 °C for 48 h. In total, 94 colonies were picked up, purified (3 times) and morphologically identified [14]. Then, biomass production was evaluated in liquid medium (per 1000 mL contained yeast extract 10 g, peptone 20 g, glucose 20 g, adenine sulfate 0.03 g) at 30 °C for 48 h based on optical density (OD) at 600 nm. Fourteen isolates with smooth round colony and OD > 1.0 were finally selected. The growth curves of the 14 isolates were further measured in liquid medium at 39 °C, 150 rmp for 48 h to evaluate their growth potential in the rumen. Triplicate was set for each isolate. OD was measured at an interval of 4 h. To identify these isolates, internal transcribed spacer (ITS) regions of fungi were sequenced and BLAST in GenBank. The No. 8 isolate belonging to *P. kudriavzevii* had the highest OD value. Therefore, isolate 8 was used to perform animal feeding experiment. A Plant/Fungi DNA Isolation Kit (Omega Bio-Tek, Shanghai, China) was used to extract the yeast DNA following the instructions. PCR amplification was performed targeting ITS using the primers ITS1F/ITS4R [18]. The sequence of primer ITS1F was 5′-TCCGTAGGTGAACCTGCGG-3′, and ITS4R was 5′-TCCTCCGCTTATTGATATGC-3. PCR was performed in 50 μL reactions including 5 × FastPfu buffer 10 μL, 2.5 mmol/L dNTPs 2 μL, each primer 1 µL, FastPfu Polymerase (AP221-01, Tran, Beijing, China) 0.5 µL. The procedure of PCR was 95 °C, 5 min; 95 °C for 30 s, 55 °C for 30 s, 72 °C for 45 s, 30 cycles; 72 °C for 10 min. The amplicons were sequenced using a 3730XL DNA Analyzer (Applied Biosystems, Foster, CA, USA).

Live yeast additive (4 × 10^8^ CFU/g) was prepared using a freeze dryer (JK-FD-1N, Shanghai Jingke Scientific Instrument Co., Ltd., Shanghai, China). Prior to freeze-drying, 10% glycerine and 12% trehalose were mixed with the culture of isolate 8, which had been incubated at 30 °C, 48 h. The procedure of freeze-drying was −50 °C, 90 min; −40 °C, 60 min; −30 °C, 60 min; −20 °C, 60 min; −10 °C, 60 min; 0 °C, 180 min; 10 °C, 300 min; 20 °C, 180 min; 30 °C, 180 min; 30 °C, 300 min. Finally, 5000 g of live yeast additive was stored under seal at 4 °C until use.

### 2.2. Animal Trial

The animal trial was carried out from January to February 2021, at a sheep farm in Huzhou, China. Randomized block design was used in this study. Sixteen healthy Hu sheep (Age (mean ± SD) = 108 ± 5 d, Body weight = 28.32 ± 0.71 kg) were randomly assigned into CON and LPK groups. Sheep assigned to CON were fed a basal ration (*n* = 8). Sheep assigned to LPK were fed a basal ration plus live yeast additive (20 g/d/head, *n* = 8). All sheep were housed individually (pen, 0.9 m × 1.5 m), and fed twice at 08:00 am and 5:00 pm every day. Free access to water was given. Approximately 5% feed residual was allowed to reduce sorting. The adaptation period was 1 week, and the treatment period was 4 weeks. All sheep were fed pelleted rations. Live yeast was first mixed with a 10 g corn meal. Then, the mixture was dressed on the top of basal diet in the LPK group. In the CON group, only 10 g of corn meal was dressed on the top of basal diet. The basal ration was formulated according to NY/T 816-2004 (Ministry of Agriculture of China, 2004), which met the requirements of 300 g/d of weight gain for sheep weighing 30 kg (Table 1).

All sheep were weighed before morning feeding on day 1 and 28 during the treatment period. The feed given and orts were recorded every day. Average daily gain (body weight gain divided by no. of days) and feed efficiency (DMI/BW gain, kg/kg) were calculated.

### 2.3. Sample Collection

Feed samples were collected weekly. Rectum contents (fecal samples) were collected on day 26, 27 and 28. The mL sulfuric acid solution (concentration, 10%) was added to each 100 g fecal sample to fix nitrogen. Finally, all fecal samples from each sheep were mixed and stored at −20 °C.

Rumen contents were sampled on day 28 using an oral tube (Wuhan Kelibo Equipment Co., Ltd., Wuhan, China) at 4 h after morning feeding. In order to avoid saliva contamination, discard the first 50 mL of rumen content [20]. A portable pH meter (HI 9024C; HANNA Instruments, Woonsocket, RI) was used to determine rumen content. pH. Five mL of rumen content was preserved at −80 °C for subsequent microbiota assays. The remaining rumen content was filtrated through four layers of gauze. The filtrate was conserved at −20 °C for further measuring VFAs, ammonia nitrogen (NH_3_-N) and microbial crude protein (MCP).

### 2.4. Chemical Analysis

The DM of feed and feces samples was measured [21]. The samples were firstly dried at 65 °C for 48 h, then dried at 105 °C for 3 h using an air-drying oven. CP was measured using a Kjeldahl apparatus (Kjeltec 8400, Shenzhen, China), with a pretreated process by copper sulfate and concentrated sulfuric acid [21]. EE was measured using a Soxhlet extractor with anhydrous ether as an extractant [21]. NDF and ADF were measured using a fiber analyzer (ANKOM200, ANKOM Technologies, Macedon, NY, USA) by washing with neutral detergent, anhydrous sodium sulfite and α-amylase, and acid detergent [22]. The total tract apparent digestibility (TAD) was measured using acid insoluble ash (AIA) as an interior label [23]. AIA was measured using a muffle furnace (Sx2-4-10N, Tianjin China) with a process treated by hydrochloric acid. The VFAs were measured using a GC-14B (Shimadzu, Shijota, Japan) [24]. Crotonic acid solution was mixed with samples as internal standard. Lactate was determined with sulfuric acid, copper sulfate and p-hydroxybiphenyl [25]. Ammonia nitrogen was determined using the indophenol method [26]. MCP was measured using a colorimetric method [27].

### 2.5. Microbial DNA Extraction, Sequencing and Data Analysis

Microbial DNA of the rumen content (200 mg) was extracted using an E.Z.N.A. Soil DNA kit (Omega Bio-Tek, USA) combining a bead-beating process. The bacterial sequencing was targeted on the V3 and V4 regions of 16S rRNA genes. The primers’ sequences were 341F (5′-CCT AYG GGR BGC ASC AG-3′) and 806R (5′-GGA CTA CNN GGG TAT CTA AT-3′) [28]. Primers for amplifying eukaryotic 18S rRNA genes included TAReukaseFWD1 (5′-CCA GCA SCY GCG GTA ATT CC-3′) and TAReukREV3 (5′-ACT TTC GTT CTT GAT YRA-3′) [29]. PCR amplicons were sequenced on an Illumina MiSeq PE 300 platform in Shanghai BIOERON Biotech. Co., Ltd. (Shanghai, China).

Paired-end reads were merged using FLASH (version 1.2.7) [30]. Then, reads were assigned into amplicon sequence variants (ASVs) at 100% identical similarity using DADA2. The representative ASV sequences were identified by SILVA (138.1) and classified into specific databases of bacteria (16S rRNA sequencing), protozoa and fungi (18S rRNA sequencing). The abundance of microbial communities was tested for significance based on the method of the non-parametric test (Mann–Whiney U) using SPSS (v. 25). Alpha diversity (Chao 1 indices, ACE indices and Shannon indices) was calculated using QIIME2 [31]. The PCA was implemented in the R program (v. 3.6.1) based on Bray–Curtis distance. The analysis of similarities (ANOSIM) was determined for the difference between the CON and LPK groups. The correlations among *P. kudriavzevii*, VFAs’ concentration and the microbial relative abundance were assessed by Spearman’s test.

### 2.6. Statistical Analysis

The data of the growth performance, digestibility and rumen fermentation were analyzed with a general linear model (single variable) by SPSS (v. 25, SPSS Inc., Chicago, IL, USA). Fixed factors were group and treatment. Covariant was the initial body weight of sheep. The data of microbes were analyzed by the nonparametric test (Kruskal–Wallis). The post hoc test was performed by independent samples test to evaluate the difference between any two groups. A significant level was indicated at *p* < 0.05.

## 3. Results

### 3.1. Growth Curve of Yeast Isolates

The growth curve of 14 yeast isolates was determined (Figure 1). From 8 to 24 h after inoculation, these isolates entered the logarithmic growth phase, then entered the stable growth phase after 32 h of inoculation. Isolate 8 had the fastest growth rate and highest OD value, and formed white smooth colonies with elliptical edges and a convex center. The sequence of the ITS region showed that isolate 8 belonged to *P. kudriavzevii.*

### 3.2. Animal Performance and Nutrient Digestibility

LPK did not affect DMI (*p* = 0.906), initial BW (*p* = 0.292), ADG (*p* = 0.906) and feed efficiency (*p* = 0.300). The final BW was significantly increased in LPK compared to CON (*p* = 0.033, Table 2). The digestibility of DM (*p* = 0.033), EE (*p* = 0.010), NDF (*p* = 0.003) and ADF (*p* = 0.005) in LPK was higher than CON, but CP was not affected (*p* = 0.121, Table 3).

### 3.3. Rumen Fermentation Parameters

The concentrations of acetate (*p* < 0.001), total VFAs (*p* = 0.005) and the acetate-to-propionate ratio (*p* = 0.011) were higher in the LPK group. However, ruminal pH (*p* = 0.618) and concentrations of NH_3_-N (*p* = 0.466), lactate (*p* = 0.988), MCP (*p* = 0.562) and other VFAs (*p* > 0.05) were not affected by treatment (Table 4).

### 3.4. Rumen Microbial Community Diversity

PCoA analysis showed that LPK did not separate the rumen bacterial communities from CON. (ANOSIM: R = 0.082, *p* = 0.174; Figure 2A). However, there were clear segregation and dissimilarities for ruminal protozoa (ANOSIM: R = 0.423, *p* = 0.001; Figure 2B) and fungi (ANOSIM: R = 0.326, *p* = 0.002; Figure 2C).

The α-diversity was shown in Table 5. For bacteria, the richness was higher in LPK than CON (Observed ASVs, *p* = 0.029; Chao 1, *p* = 0.030). However, LPK did not affect the Shannon (*p* = 0.110) and Simpson index (*p* = 0.213). For eukaryotes, LPK did not affect the α-diversity.

### 3.5. Rumen Microbiota Composition

The dominant phyla were Bacteroidetes, Firmicutes and Proteobacteria in the two groups, accounting for about 93% to 97% of the total phyla, respectively. Thirty dominant genera were > 0.5% in proportion. The first three dominant bacterial genera were *Ruminococcus*, *Selenomonas* and *Prevotella*. The relative abundance of *Anaerovibrio* (*p* = 0.028), *Pseudomonas* (*p* = 0.028), *Faecalibacterium* (*p* = 0.001) and *Bifidobacterium* (*p* = 0.038) was higher in LPK than CON. While the percentages of *Pseudoscardovia* (*p* = 0.011) and *Syntrophococcus* (*p* = 0.035) were lower in LPK (Figure 3B).

All protozoa identified belonged to the phylum Ciliophora. At the genus level, Entodinium was the most abundant genus, accounting for 60.25% and 77.14% in LPK and CON, respectively. The relative abundances of *Polyplastron* (*p* = 0.002) and unclassified Trichostomatia (*p* = 0.004) were higher in LPK than CON, while the percentage of unclassified Hypotrichia was lower in LPK (*p* = 0.032, Figure 4B).

For the rumen fungi, the dominant phyla were Ascomycota and Basidiomycota, accounting for more than 99% of the total phyla. The relative abundance of Ascomycota was higher in LPK than CON (*p* = 0.003; Figure 5A), while Basidiomycota was lower in LPK (*p* = 0.003; Figure 5A). The dominant genera (the relative abundance > 1%) were *Kurtzmaniella Candidaclade*, *Pichia* and the *Clavispora-Candida clade*. The relative abundance of *Pichia* was higher in LPK than in CON (*p* = 0.012; Figure 5B).

### 3.6. Correlation Analysis

Spearman correlation analysis was performed among *P*. *kudriavzevii*, rumen VFAs and microorganisms (the top 10 bacteria, fungi and protozoa based on relative abundance at genus level). For the bacteria (Figure 6), the abundance of *Prevotella* was negatively correlated with the concentration of propionate (R = −0.65; *p* = 0.007) and positively correlated with the acetate-to-propionate ratio (R = 0.676; *p* = 0.005). *Selenomonas* (R = 0.673; *p* = 0.005) and *Anaerovibrio* (R = 0.682; *p* = 0.004) were positively correlated with the concentration of total VFA. Unclassified Clostridia UCG-014 had a positive correlation with butyrate concentration (R = 0.597; *p* = 0.016). For the protozoa, *Polyplastron* (R = 0.809; *p* < 0.001) was positively correlated with *P*. *kudriavzevii* and had a positive correlation with acetate concentration (R = 0.567; *p* = 0.022) and acetate-to-propionate ratio (R = 0.547; *p* = 0.028). *Oxytricha* was positively associated with the concentration of valerate (R = 0.199; *p* = 0.049), and *Acineta* was negatively associated with the acetate-to-propionate ratio (R = −0.524; *p* = 0.037). For the fungi, *P*. *kudriavzevii* showed a positive correlation with Pichia (R = 0.924; *p* < 0.001) and negatively correlated with *Malassezia* (R = −0.668; *p* = 0.005), whereas *Pichia* was positively linked with the acetate concentration (R = 0.661; *p* = 0.005) and acetate-to-propionate ratio (R = 0.595; *p* = 0.015). *Candida* was negatively correlated with the concentration of Valerate (R = −0.571; *p* = 0.020). There was a negative correlation between the *Kazachstania-Candida clade* and the Propionate concentration (R = −0.508; *p* = 0.044) (Figure 6).

## 4. Discussion

In the present study, the function of rumen native yeast *P. kudriavzevii* was investigated through animal *in vivo* and *in vitro* experiments. It had been reported that the growth of yeast was affected by species, oxygen, nutrients, temperature and other aspects [32]. Under a healthy physiological state, the rumen temperature is maintained at 38.5 to 40 °C. Therefore, the biomass production capability of the obtained isolates was evaluated at 39 °C in the present study. Isolate 8 grew rapidly and had a relatively high OD value, suggesting that this isolate would grow faster and have greater biomass in rumen conditions. It was identical to the previous report by Suntara et al. [15]. Therefore, we predicted that isolate 8 (*Pichia kudriavzevii*) would maintain activity in the rumen environment and regulate rumen microbial composition.

Isolate 8 elevated the concentrations of acetate and total VFA, which were consistent with previous studies [33,34]. Since VFA produced by microbial fermentation on carbohydrates in the rumen provides more than 70% of the host’s energy requirements [35], a higher concentration of total VFA would be expected to provide more energy to the body. An increased concentration of acetate may hint at an enhanced ability to digest structural carbohydrates [36]. This observation corresponded to the increase in nutrient digestibility. Isolate 8 feeding increased the digestibility of nutrients, which was identical to the previous studies on other yeast products, such as *Saccharomyces cerevisiae* [34,37]. Therefore, isolate 8 feeding may promote fiber degradation and acetate fermentation by regulating rumen microbiota.

To further explore the effects of isolate 8 on the rumen microbiota, we focused on the bacterial, protozoal and fungal communities, which were the main fermentation microbial populations. Isolate 8 feeding increased bacterial richness, which was consistent with the studies using commercial *S. cerevisiae* on dairy cows [38]. Isolate 8 increased the relative abundance of *Anaerovibrio*, which plays an important role in rumen lipid degradation [39]. By producing extracellular lipase, *Anaerovibrio* decomposes lipids and glycerol into free VFAs [40,41]. This result may explain the increased digestibility of EE by isolate 8. The biomass of *P. kudriavzevii* is rich in glycerol [42], which might be used as a substrate to promote the growth of *Anaerovibrio*. Our data showed that isolate 8 feeding increased the relative abundance of *Pseudomonas,* which can perform functions including hydrocarbon degradation, xenobiotic degradation, nitrification, denitrification, cellulose degradation and lipolysis [43,44,45]. The higher proportion of *Pseudomonas* may improve feed digestibility in Hu sheep. In addition, isolate 8 feeding also promoted the growth of two probiotics, *Faecalibacterium* and *Bifidobacterium*. *Faecalibacterium* can reduce the severity of inflammation and enhance intestinal barrier function by releasing metabolites [46], while *Bifidobacterium* can stimulate and maintain the intestinal mucosal barrier and immune response and also produce a series of beneficial metabolic substrates to prevent the attachment of pathogens, which exerts positive effects in the intestinal tract of young animals [47]. An improvement in the relative abundance of *Bifidobacterium* can reduce the risk of disease and improve animal performance [48]. Taken together, these findings showed that isolate 8 feeding improved ruminal digestion of lipids and increases the abundance of probiotics.

Tripathi and Karim reported that feeding yeast increased rumen ciliates [49]; although, some studies also have shown the opposite results [50]. In the current study, isolate 8 feeding promoted the growth of the dominant genera of protozoa, *Polyplastron* and unclassified Trichostomatia, and decreased the relative abundance of unclassified Hypotrichia. As common rumen protozoa, *Polyplastron* is capable of producing cellulase and xylanase to decompose structural polysaccharides such as cellulose and hemicellulose [51], contributing to the degradation of fibers in the LPK group. Therefore, our study suggested that isolate 8 enhanced the relative abundance of ruminal-specific protozoa to promote fiber degradation.

Only a few studies, to date, have investigated the effects of yeast on the community of rumen fungi [16]. Chaucheyras-Durand believed that the addition of yeast can promote the fixation of fungi in feed pellets and thus affect the digestive degradation of fiber in the rumen [52]. Isolate 8 increased the relative abundance of Ascomycota and decreased Basidiomycota. Isolate 8 also increased the relative abundance of *Pichia*. The increase in *p. kudriavzevii* was the crucial factor for the increase in *Pichia*. As a fiber-degrading fungus, *P. kudriavzevii* can help the host to degrade cellulose, which was identical to the results of the correlation analysis. *Pichia* was positively correlated with the acetate concentration and acetate-to-propionate ratio. Even though the activity of isolate 8 was not measured in the rumen of Hu sheep, taking all the results into consideration, isolate 8 might have a high potential to survive and exert beneficial effects through interacting with native microbiota in the rumen [4]. In addition, the high biomass of *P. kudriavzevii* might provide rumen microorganisms with rich nutrients such as organic acids, peptides and amino acids [3].

## 5. Conclusions

A *P. kudriavzevii* isolate was obtained from the rumen of sheep. Live *P. kudriavzevii* feeding promoted the performance of rumen acetate-type fermentation and digestibility of nutrients in Hu sheep. *P. kudriavzevi* enhanced specific ruminal microbial populations, including lipolytic bacteria (e.g., *Anaerovibrio* spp. and *Pseudomonas* spp.), probiotic bacteria (e.g., *Faecalibacterium* spp. and *Bifidobacterium* spp.) and fiber-degrading eukaryotes (e.g., *Pichia* spp. and *Polyplastron* spp.). Therefore, rumen native yeast *P. kudriavzevii* has a high potential for use in sheep production.

## Figures and Tables

**Figure 1 jof-08-01260-f001:**
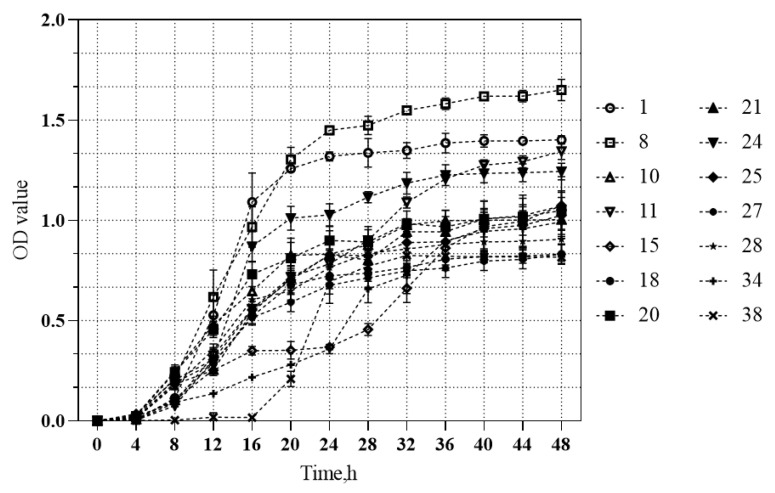
Growth curves of 14 rumen-derived yeast strains incubated for 48 h at 39 °C and 150 rmp in a culture-shaker under aerobic conditions.

**Figure 2 jof-08-01260-f002:**
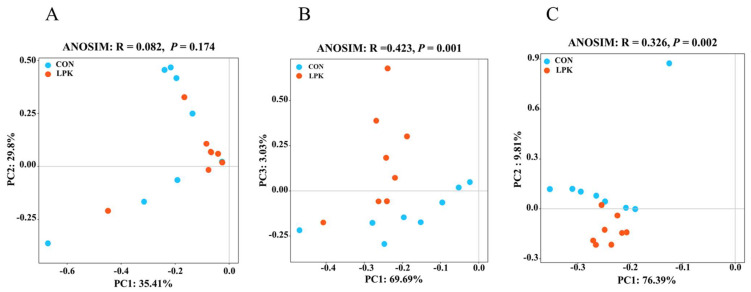
Principal coordinate analysis (PCoA) of rumen bacteria (**A**), protozoa (**B**), and fungi (**C**) based on operational taxon data. Control (CON) group (blue circle), live *Pichia kudriavzevii* (LPK) group (red circle).

**Figure 3 jof-08-01260-f003:**
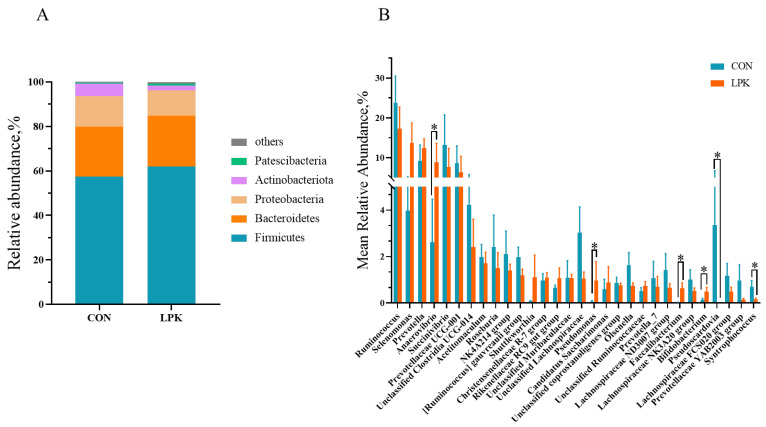
The bacterial community in the rumen of Hu sheep: (**A**) The mean relative abundance of phylum ≥ 0.5% in at least one group is present. (**B**) The mean relative abundance of genus ≥ 0.5% in at least one group is present. Error bars, SEM. *, *p* < 0.05. CON = control group; LPK = live *Pichia kudriavzevii* group.

**Figure 4 jof-08-01260-f004:**
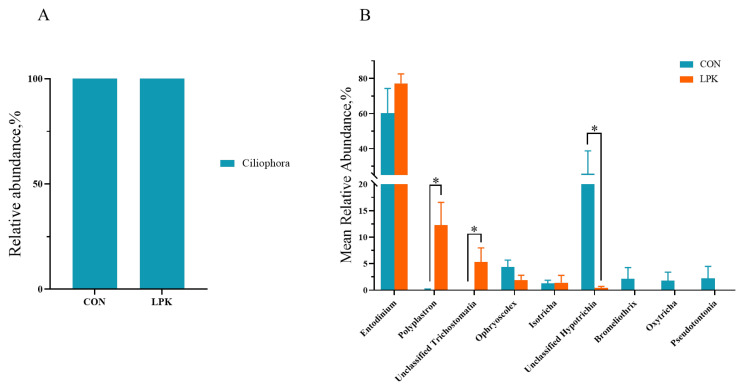
The rumen protozoa community in Hu sheep: (**A**) The mean relative abundance of phylum ≥ 1% in at least one group is present. (**B**) The mean relative abundance of genus ≥ 1% in at least one group is present. Error bars, SEM. * *p* < 0.05. CON = control group; LPK = live *Pichia kudriavzevii* group.

**Figure 5 jof-08-01260-f005:**
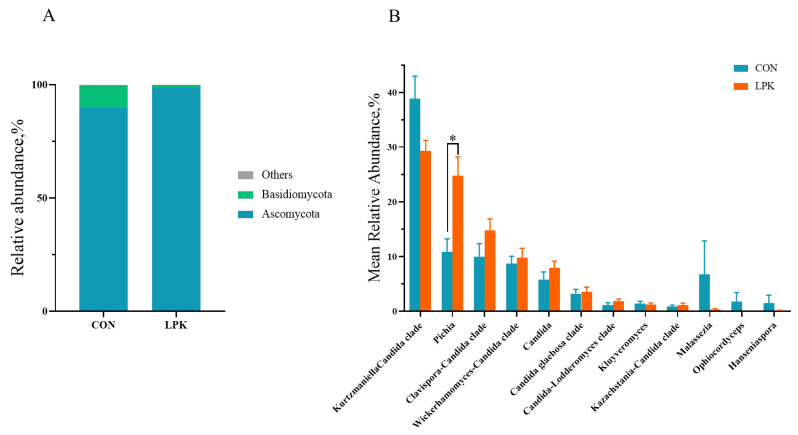
The rumen fungal community in Hu sheep: (**A**) The mean relative abundance of phylum ≥ 1% in at least one group is present. (**B**) The mean relative abundance of genus ≥ 1% in at least one group is present. Error bars, SEM. * *p* < 0.05. CON = control group; LPK = live *Pichia kudriavzevii* group.

**Figure 6 jof-08-01260-f006:**
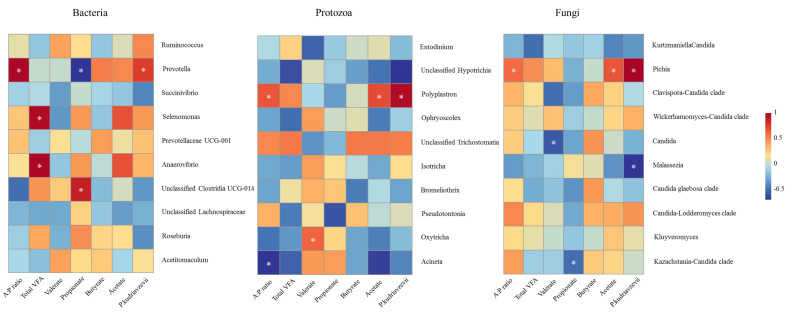
Spearman rank correlation analysis between rumen microbial community (top 10 relative abundance at the genus level of bacteria, protozoa and fungi), *Pichia kudriavzevii* and rumen volatile acids. Red, positive; blue, negative. * *p* < 0.05.

**Table 1 jof-08-01260-t001:** Composition and nutrient levels of diets.

Items	Content
Ingredient (air-dried basis)	
Mushroom residue ^1^	10.00
Maize	40.00
Soybean meal	10.00
Corn germ meal	10.00
Barley malt sprouts	5.50
Barley peel	10.00
Rice husk	6.00
Rice bran	3.50
NaCl	0.80
Dicalcium phosphate	0.80
Limestone meal	1.40
Mineral and vitamin mixture ^2^	2.00
Total	100.00
Nutrient level (dry matter basis)	-
Metabolic energy (ME) ^3^ (MJ·kg^−1^)	11.01
Dry matter (DM), %	87.42
Crude protein (CP), %	14.00
Ether extract (EE), %	3.12
Neutral detergent fiber (NDF), %	43.16
Acid detergent fiber (ADF), %	23.55
Crude ash (Ash), %	9.54
Calcium (Ca), %	1.30
Phosphorus (P), %	0.40

^1^*Flammulina velutipes* residue: the remaining cultivation substrate after the harvest of *F. velutipes*, containing moisture 5.59%, CP 12.7%, NDF 45.2%, ADF 22.3%, EE 1.8%, and Ash 11.6%. ^2^ Mineral and vitamin mixture was composed of Fe, 3.5 g; Zn, 4.5 g; Cu, 2.0 g; Mn, 3.0 g; Co, 80 mg; I, 140 mg; Se, 40 mg; vitamin A, 550,000 IU; vitamin D, 48,000 IU; vitamin E, 2000 IU. ^3^ ME was the calculated value [19].

**Table 2 jof-08-01260-t002:** Effects of live *Pichia kudriavzevii* on growth performance of Hu sheep ^1^.

Items	CON	LPK	*p*-Value
Dry matter intake, g/d	1519.06 ± 21.95	1551.13 ± 23.29	0.906
Initial body weight, kg	27.97 ± 0.99	28.66 ± 1.08	0.292
Final body weight, kg	34.90 ± 1.01	36.23 ± 0.93	0.033
Average daily gain, g/d	241.33 ± 19.31	266.83 ± 14.81	0.906
Feed efficiency	6.07 ± 0.45	5.81 ± 0.50	0.300

^1^ The values are present as means ± SEM.

**Table 3 jof-08-01260-t003:** Effects of live *Pichia kudriavzevii* on nutrient digestibility of Hu sheep ^1^ (%).

Items	CON	LPK	*p*-Value
Dry matter	60.62 ± 0.37	64.41 ± 1.46	0.033
Crude protein	60.50 ± 1.85	64.68 ± 1.16	0.121
Ether extract	58.08 ± 2.65	67.56 ± 1.41	0.010
Neutral detergent fiber	44.38 ± 1.26	51.44 ± 1.28	0.003
Acid detergent fiber	37.65 ± 1.35	45.5 ± 1.63	0.005

^1^ The values are present as means ± SEM.

**Table 4 jof-08-01260-t004:** Effects of live *Pichia kudriavzevii* on rumen fermentation parameters of Hu sheep ^1^.

Items	CON	LPK	*p*-Value
Rumen pH	6.30 ± 0.08	6.18 ± 0.39	0.618
Ammonia, mg/dL	19.23 ± 1.97	21.68 ± 2.27	0.466
Microbial crud protein, mg/dL	43.55 ± 0.04	38.69 ± 0.06	0.562
Lactate, mmol/L	0.26 ± 0.02	0.26 ± 0.02	0.988
Acetate, mmol/L	40.73 ± 1.67	52.08 ± 2.02	<0.001
Propionate, mmol/L	24.42 ± 2.15	20.94 ± 0.91	0.137
Isobutyrate, mmol/L	0.17 ± 0.05	0.29 ± 0.05	0.098
Butyrate, mmol/L	16.27 ± 1.26	17.36 ± 1.03	0.444
Isovalerate, mmol/L	0.47 ± 0.08	0.47 ± 0.05	0.997
Valerate, mmol/L	1.77 ± 0.20	1.56 ± 0.15	0.382
Acetate/propionate	1.76 ± 0.19	2.52 ± 0.14	0.011
Total volatile fatty acids, mmol/L	83.84 ± 2.58	92.69 ± 1.72	0.005

^1^ The values are present as means ± SEM.

**Table 5 jof-08-01260-t005:** Effects of live *Pichia kudriavzevii* on microbial community diversity in the ruminal content of Hu sheep ^1^.

Item	CON	LPK	*p*-Value
Rumen bacteria			
Observed ASVs ^2^	167.38 ± 17.58	230.75 ± 19.12	0.029
Coverage	1.00 ± 0.0002	1.00 ± 0.0002	0.255
Chao 1	170.38 ± 18.50	234.25 ± 18.93	0.030
Shannon	3.24 ± 0.26	3.80 ± 0.20	0.110
Simpson	0.15 ± 0.04	0.09 ± 0.02	0.213
Rumen protozoa			
Observed ASVs	11.00 ± 2.30	10.50 ± 1.09	0.847
Coverage	0.98 ± 0.01	0.98 ± 0.003	0.618
Chao 1	13.13 ± 3.01	11.38 ± 1.12	0.600
Shannon	1.41 ± 0.30	1.52 ± 0.16	0.740
Simpson	0.43 ± 0.11	0.34 ± 0.06	0.512
Rumen fungi			
Observed ASVs	24.38 ± 3.18	23.38 ± 1.53	0.781
Coverage	0.98 ± 0.01	0.98 ± 0.003	0.868
Chao 1	28.00 ± 4.53	29.00 ± 4.74	0.881
Shannon	2.20 ± 0.13	2.28 ± 0.08	0.597
Simpson	0.20 ± 0.03	0.16 ± 0.01	0.148

^1^ The values are presented as means ± SEM; ^2^ ASV_S_ = Amplicon Sequence Variants.

## Data Availability

The raw reads of the 16S rRNA gene and 18S rRNA gene sequencing obtained by our research are accessible under the Sequence Read Archive of the NCBI with the accession numbers PRJNA896857.

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
