# Peer review of "Isolation and Characterization of Ruminal Yeast Strain with Probiotic Potential and Its Effects on Growth Performance, Nutrients Digestibility, Rumen Fermentation and Microbiota of Hu Sheep"

_jof, 2022, doi:10.3390/jof8121260_

Round 1

Reviewer 2 Report

This study isolated and characterized a rumen native yeast, and investigated its effects on growth performance, nutrient digestibility, rumen fermentation, and rumen bacteria, protozoa and fungi populations in Hu sheep. Rumen inhabits not only anaerobic microbes, but also facultative anaerobes, especially yeast. Yeast products have been widely used in ruminants industry, however, the effects of these yeast products vary greatly. One of the problems might be the adaptation to rumen environment. Therefore, the study of rumen native yeast is of great significance for ruminant production. There are some minor problems. 

1.     Line 20, relative abundance

2.     Line 76, please give a detail of the medium, such as components, manufacturer etc.

3.     Line 77, 94 single colonies were picked up.

4.     Line 78, use “Fourteen” to replace “14”

5.     Line 87-90, give more detail for the PCR amplification and ITS sequencing, such as procedures, references

6.     Line 92, viable bacterial powders?

7.     Suggest to reorganize section 2.1

8.     Line 97, how many grams or CFU was added to per sheep per day

9.     Line 113 and 133, sample collection?

10.  Line 133,153,173, “2.4.”?

11.  Table 1, what is mushroom residue? Which kind of mushroom?

12.  Line 207, discuss the decline of dry matter intake in LPK group

Reviewer 3 Report

This article is a complete story, and the paper was well organized and written. The fungi was screened from the rumen and compared and evaluated in Hu sheep, and may be used as a market product in the future. However, the article has the following minor defects:

1.The L225 title is wrong;

2. What is the role of rice husk in Table 1? And since NDF and ADF already exist, CF is unnecessary;

3. This is also the most important point I think. How do you determine your addition dosage of L97? What is the basis? Did you do the pre experiment before this formal experiment? Why is this dose so effective? 

Round 2

Reviewer 1 Report

Thank you for responding to the comments.

I see that you responded to most comments, but you need to clarify the freeze-drying method. Also, please provide a clean manuscript to  read it.

Thank you

Author Response

Thanks for this comment. More information has been added."Live yeast additive (4 × 108 CFU/g) was prepared using a freeze dryer (JK-FD-1N, Shanghai Jingke Scientific Instrument Co., Ltd., Shanghai, China). Prior to freeze-drying, 10% glycerine and 12% trehalose were mixed with the culture of isolate 8, which had been incubated at 30 °C, 48 h. The procedure of freeze-drying was -50 °C, 90 min; -40 °C, 60 min; -30 °C, 60 min; -20 °C, 60 min; -10 °C, 60 min; 0 °C, 180 min; 10 °C, 300 min; 20 °C, 180 min; 30 °C, 180 min; 30 °C, 300 min. Finally, 5000 g of live yeast additive was stored under seal at 4 °C until use."(Page 3, Line 99-105)